# The Representation of Orientation Semantics in Visual Sensory Memory

**DOI:** 10.3390/bs15010001

**Published:** 2024-12-24

**Authors:** Jingjing Hu, Xutao Zheng, Haokui Xu

**Affiliations:** 1School of Education, Zhejiang International Studies University, Hangzhou 310028, China; hujj@zisu.edu.cn; 2Department of Psychology and Behavior Sciences, Zhejiang University, Hangzhou 310023, China; 3Department of Psychology, College of Education, Zhejiang University of Technology, Hangzhou 310028, China

**Keywords:** sensory memory, visual representation, orientation, masking effect

## Abstract

Visual sensory memory constructs representations of the physical information of visual objects. However, few studies have investigated whether abstract information, such as semantic information, is also involved in these representations. This study utilized a masking technique combined with the partial report paradigm to examine whether visual sensory memory representation contains semantic information. Here, we regarded the concept of orientation carried by the visual stimulus as semantic information. In three experiments, participants were asked to remember the orientation of arrows. Visual stimuli with orientation information (triangles, rectangles, and Chinese characters) and without orientation information (circles, squares, and different Chinese characters) were used as masks. The results showed that memory performance was worse when masks contained orientation information compared to when they did not, as similar orientation semantic information between masks and targets created visual representation conflicts. These findings suggest that visual sensory memory representation includes the semantic information of orientation.

## 1. Introduction

Cognitive systems with limited resources understand and interact with the complex environment by selectively processing information ([2]; [20]). However, external information frequently changes, making it challenging to select relevant data during brief presentation periods. Consequently, the visual system must temporarily store large amounts of information, relying on sensory memory ([5]; [30]). Sensory memory, while possessing a much higher storage capacity than visual short-term memory, is relatively short-lived and its contents rapidly decay ([34]). Through the selection mechanisms of the visual system, a portion of sensory memory is consolidated into short-term memory ([13]).

Visual representation is a central focus in sensory memory research. Early studies suggested that visual sensory memory forms a representation of the physical world as a snapshot ([8]; [35]). This snapshot was not a strict replication of the physical world but resembled a mental image that included key physical features, allowing for flexible manipulation by the visual system. For instance, research using masking techniques found that sequential presentation of multiple visual stimuli can result in fusion or overlay, with short intervals leading to fusion and long intervals producing overlay ([9]; [27]). Even a virtual geometric shape used as a mask can interfere with the memory of the target stimulus ([9]). Further research showed that when the target stimulus (e.g., Arabic numerals “2856”) was similar to the mask stimulus (e.g., Arabic numerals “3491”), participants’ memory performance for the target decreased significantly ([4]; [25]). Researchers concluded that this decline in memory performance likely arises from a conflict between the visual representations of the target and mask stimuli, implying that these representations contain specific physical feature information.

The similarity of physical features is not the only factor explaining the masking effect. Although the mask and the target were distinct in identity (e.g., different Arabic numerals), they shared the same semantic information (e.g., the concept of numbers). This suggests that conflicts between target and mask representations may also arise from shared semantic content. Although early theories emphasized that early visual processing is pre-categorical and that sensory memory does not involve any concepts or semantics ([6]; [7]; [11]), contemporary theories no longer adhere to these views ([16]). For instance, the dual-trace model of memory posits that sensory memory occurring within 500 ms contains dual representations, both of which include the extraction of scene meaning ([5]). This perspective is supported by numerous studies showing that the human visual system can recognize high-level information, like semantic concepts (e.g., subitizing), from simple visual stimuli ([3]; [17]; [21]; [28]; [22]). Even infants and young children understand the meaning and intentions behind visual stimuli ([26]; [39]). Moreover, visual selection depends not only on physical features but also on high-level aspects, such as semantic information ([24]; [29]), which means that high-level features must be incorporated into sensory memory representation before information selection occurs. Recently, Hu et al. found that in a sensory memory task using Arabic numerals as targets, applying Chinese capital numerals as masks induced a stronger masking effect than either a noise mask or an inverted Chinese character mask ([18]). Given the clear physical differences between Chinese capital and Arabic numerals, the stronger masking effect likely stemmed from the semantic similarity between the target and mask, suggesting that sensory memory’s visual representation contains semantic information.

The masking effect of Chinese capital numerals on Arabic numerals provided direct evidence for the presence of semantic information in sensory memory representations. However, it remains unclear whether this effect is a general characteristic of sensory memory representation. Both Arabic numerals and Chinese capital numerals are symbols that convey quantitative information, which constitutes a type of semantic information ([28]; [18]). When the task involves recalling and identifying Arabic numerals, it inherently requires the representation of semantic information. Yet, does the representation retain semantic information when the memory task involves only the physical features of the target?

The current study uses the orientation feature to investigate the above question. Orientation, a typical physical feature of visual objects, is widely used in vision research ([12]; [14]). Research has shown that specialized neurons in the primary visual cortex process orientation features ([19]; [23]). Thus, the task of memorizing orientation is generally considered to involve only the processing of basic physical features. In this study, we will use objects with orientation information (e.g., arrows) as targets and objects with or without orientation information but shared similar physical features (e.g., triangles and circles) as masks. By comparing the interference caused by the two types of masks, we can determine whether the representation contains semantic information about orientation. If the mask with orientation information causes stronger interference compared with the mask with similar physical features but no orientation information, it would indicate a conflict between the semantic orientation information of the mask and that of the target. This would suggest that sensory memory representation contains orientation-related semantic information.

## 2. Experiment 1

### 2.1. Method

#### 2.1.1. Participants

A priori power analysis was conducted using G*Power 3.1. There were two within-subject factors, with a primary focus on the factor of mask type, which had two levels. The study design was consistent with that of Hu et al. ([18]), referencing the experiment with the smallest effect size (Experiment 3) to determine the expected effect size ($\eta_{p}^{2}$ = 0.170) and significance level (α = 0.05). A sample size of 18 participants was sufficient to achieve a power of 0.95. To balance the order of all conditions across participants, we ultimately recruited 24 undergraduate and graduate students (17 females and 7 males, aged 18 to 24 years) from Zhejiang University. Participants received compensation in the form of credits or money. All participants had normal or corrected-to-normal vision and provided informed consent (the naked eye or with glasses was above 4.8 as tested by international standard vision chart). The study was approved by the Institutional Review Board at the Department of Psychology and Behavioral Sciences, Zhejiang University.

#### 2.1.2. Apparatus and Stimuli

The stimuli and program were developed using MATLAB and Psychtoolbox 3.0 and presented on a 19-inch CRT monitor (DELL P992) with a spatial resolution of 1280 × 962 pixels and an 85 Hz refresh rate. Participants maintained a visual distance of approximately 60 cm from the center of the monitor. The experiment was conducted in a closed, dimly lit room, with the primary light source being the display screen. Participants provided responses via the computer keyboard.

The targets were 12 arrows arranged in a 3 × 4 matrix in the center of the screen. Each arrow was randomly generated from 8 directions (0°, 45°, 90°, 135°, 180°, 225°, 270°, 315°). At a visual distance of 60 cm, the height of the arrow was 0.60° (0.63 cm) and the interval between arrows was 0.56° (0.59 cm).

Two types of masks were used: (a) a circle with a diameter matching the height of the target arrow, and (b) a triangle with a base matching the arrow’s width and a height from the vertex to the base matching the arrow’s height (Figure 1a). Note that we adjusted the angle of the triangle mask at each position to differ from the arrow angle at the corresponding position. This adjustment aimed to investigate the effects of the semantic of the concept of orientation rather than specific orientation similarity. All stimuli were presented within a 5° visual field. The masks were positioned to fully cover the corresponding targets.

All stimuli were presented on a gray background (RGB: 41, 41, 41; brightness: 16.0 $cd/m^{2}$), with brightness contrasts of 50% for targets and 100% for masks relative to the background, ensuring sufficient coverage by the masks.

#### 2.1.3. Procedures and Design

At the start of each trial, a fixation point appeared in the center of the screen. After 1.4 s, the fixation point disappeared, followed by the target presentation for 23.5 ms (2 frames). Once the target disappeared, a blank screen appeared for 105.9 ms (9 frames), after which the masks were displayed for 23.5 ms (2 frames). In the partial report conditions, a prompt tone was presented after the masking stimulus disappeared; no prompt tone was presented in the whole report condition. In the partial report conditions, the cues were 500 ms tones, with a 2.5 kHz high tone prompting participants to report the first row of targets, an 850 Hz mid-tone for the second row, and a 200 Hz low tone for the third row (Figure 1b).

Participants were instructed to report the direction of the arrows from left to right and top to bottom. The corresponding directions were marked on the numeric keypad on the right side of the keyboard as follows (Figure 1a): 8 (0°), 9 (45°), 6 (90°), 3 (135°), 2 (180°), 1 (225°), 4 (270°), and 7 (315°). The arrows entered by the participants were displayed on the screen, allowing them to use the delete key to make corrections if needed. Participants were asked to recall as accurately as possible and were encouraged to guess when uncertain. Only after entering a sufficient number of responses (4 in the partial reporting condition and 12 in the whole report condition) could they press the Enter key to confirm and proceed to the next trial.

The experiment included two within-subject factors: mask type and report type. Mask type had two levels: circle and triangle masks. Report type had three levels: partial report with a cue delay of 0 ms, partial report with cue delay of 447 ms (38 frames), and whole report. The mask types were divided into two sections, with order balanced across participants. Each section included three blocks corresponding to the three report types, with the block order balanced across participants using a Latin square design. The experiment consisted of 280 trials in total, with 20 trials for each whole report block and 60 trials for each partial report block.

#### 2.1.4. Data Analysis

We recorded the arrow directions reported by participants for each trial and converted them into the correct recall number. In the whole report condition, the correct recall number equaled the number of correct reports made by participants. In the partial report condition, the correct recall number was three times the number of correct reports made by the participants. Repeated-measures ANOVAs were used for all statistical analyses. Greenhouse–Geisser-corrected *p*-values were reported when sphericity was violated, and Bonferroni correction was applied in further comparisons.

### 2.2. Results

We conducted a 2 (mask type) × 3 (report type) repeated-measures ANOVA. The results revealed a significant main effect of mask type (*F*(1, 23) = 54.54, *p* < 0.001, $\eta_{p}^{2}$ = 0.703), while neither the main effect of report type (*F*(2, 46) = 1.63, *p* = 0.207, $\eta_{p}^{2}$ = 0.066) nor the interaction effect (*F*(2, 46) = 0.60, *p* = 0.553, $\eta_{p}^{2}$ = 0.025) reached significance. Further comparisons showed that across all report types, participants’ performance was worse in the triangle mask condition than in the circle mask condition (*p* < 0.001 for all three report types; see Figure 2a). Moreover, the difference between correct recall number was not due to the trade-off between response time (RT) and accuracy, since the repeated-measures ANOVA for RT revealed that no significant difference in the main effect of mask type was found (*F*(1, 23) = 1.93, *p* = 0.178, $\eta_{p}^{2}$ = 0.077). These findings suggested that the triangle mask, which carried orientation semantics, interfered more strongly with sensory memory than the circle mask, which lacks orientation semantics, indicating that sensory memory representation of arrows contained semantic information about orientation.

## 3. Experiment 2

In Experiment 1, the two mask types—a circle and a triangle—differed not only in the presence of orientation information but also in their physical features. The triangle, being a polygon with distinct edges and corners, visually resembles an arrow, whereas the circle lacks such features. Consequently, the observed differences in Experiment 1 may have resulted from these physical distinctions between the two masks. To control for this factor, Experiment 2 introduced the square and the rectangle as masks. Both masks had similar physical features that were distinct from an arrow; however, the rectangle contained orientation information, whereas the square did not. If memory performance was poorer under the rectangle mask condition compared to the square mask condition, this would support the existence of an abstract representation of orientation semantics in visual sensory memory.

### 3.1. Method

#### 3.1.1. Participants

The sample size was same as that in Experiment 1. A new group of 24 undergraduate students (19 females and 5 males, age from 17 to 21 years) at Zhejiang University participated in this experiment for credit or monetary payment. All participants had normal or corrected-to-normal vision and signed informed consent. The study was approved by the Institutional Review Board at the Department of Psychology and Behavioral Sciences of Zhejiang University.

#### 3.1.2. Apparatus and Stimuli

The apparatus and stimuli used in this experiment were the same as those in Experiment 1, except there were two new masks: (a) a square whose side length was consistent with the height of the target arrow, and (b) a rectangle whose long side length was consistent with the height of the arrow and whose short side length matched the width of the arrow. Note that we adjusted the angle of the rectangle mask at each position to differ from the arrow angle at the corresponding position. The procedures, design, and data analysis methods were same as those in Experiment 1.

### 3.2. Results

A 2 (mask type) × 3 (report type) repeated-measures ANOVA was conducted. The results revealed a significant main effect of mask type (*F*(1, 23) = 11.41, *p* = 0.003, $\eta_{p}^{2}$ = 0.332). Neither the main effect of report type (*F*(2, 46) = 0.01, *p* = 0.987, $\eta_{p}^{2}$ = 0.001) nor the interaction effect (*F*(2, 46) = 0.46, *p* = 0.640, $\eta_{p}^{2}$ = 0.017) reached significance. Further comparison showed that in all report types, participants recalled worse in the rectangle mask condition than in the square mask condition (with Bonferroni correction: *p* = 0.032 for partial report with cue delay of 0 ms; *p* = 0.056 for partial report with cue delay of 447 ms; *p* = 0.005 for whole report; see Figure 2b). Moreover, the difference between correct recall number was not due to the RT-accuracy trade-off, since the repeated measures ANOVA for RT revealed that no significant difference in the main effect of mask type was found (*F*(1, 23) = 0.03, *p* = 0.869, $\eta_{p}^{2}$ = 0.001). These results showed that the rectangle mask with orientation semantics interfered more strongly in sensory memory than the square mask without orientation semantics. Given that other appearance features of the rectangle and square are highly similar, these results provided stronger evidence that representation in sensory memory contain orientation semantic information.

## 4. Experiment 3

Experiments 1 and 2 both indicated that masks with orientation information caused stronger interference than masks without orientation information, suggesting that the masking effect differences may stem from the semantic information of orientation. However, alternative explanations remain. For one, while geometric shapes and arrows appear distinct, geometric shapes with orientation information inherently share some physical features with arrows, such as marked differences in length and width. These shared features define the orientation information of both geometric shapes and arrows, making it challenging to completely isolate the influence of such physical features when using geometric shapes as a mask. Additionally, physical differences still exist between the two masks; for instance, the square has a larger coverage area than the rectangle. Given that the orientation information of geometric shapes is closely tied to their physical features, it is difficult to select two geometric masks differing in orientation semantic information while maintaining identical physical features. To mitigate these factors, Experiment 3 used Chinese characters as masks. The orientation semantic information carried by Chinese characters is clearly independent of their physical features. Therefore, if the results of the previous two experiments were replicated, it would confirm that orientation semantic information accounts for the differences in masking effect.

### 4.1. Method

#### 4.1.1. Participants

The sample size was same as that in Experiment 1. A new group of 24 undergraduate and graduate students (9 females and 15 males, age from 20 to 29 years) at Zhejiang University participated in this experiment for credit or monetary payment. All participants had normal or corrected-to-normal vision and signed informed consent. The study was approved by the Institutional Review Board at the Department of Psychology and Behavioral Sciences of Zhejiang University.

#### 4.1.2. Apparatus and Stimuli

The apparatus and stimuli used in this experiment were the same as those in Experiment 1, except there were two new masks: (a) Chinese characters without orientation semantic information, and (b) Chinese characters with orientation semantic information. These Chinese characters were presented in the font of Songti. Their size was the same as the target, so they could completely cover the corresponding target. The two types of Chinese characters have the same number of strokes and using frequency (Figure 3a). The procedures, design, and data analysis methods were same as those in Experiment 1.

### 4.2. Results

A 2 (mask type) × 3 (report type) repeated-measures ANOVA was conducted. The results revealed a significant main effect of mask type (*F*(1, 23) = 16.61, *p* < 0.001, $\eta_{p}^{2}$ = 0.419) and a significant main effect of report type (*F*(2, 46) = 3.44, *p* = 0.041, $\eta_{p}^{2}$ = 0.130). No significant interaction effect (*F*(2, 46) = 0.14, *p* = 0.870, $\eta_{p}^{2}$ = 0.006) was found. Further comparison showed that in all report types, participants recalled worse in the mask with orientation semantic information than in the mask without orientation semantic information (with Bonferroni correction: *p* = 0.049 for partial report with cue delay of 0 ms; *p* = 0.001 for partial report with cue delay of 447 ms; *p* = 0.029 for whole report; see Figure 3b). Moreover, the difference between correct recall number was not due to the RT-accuracy trade-off, since the repeated measures ANOVA for RT revealed that no significant difference of main effect of mask type was found (*F*(1, 23) = 0.10, *p* = 0.761, $\eta_{p}^{2}$ = 0.004).

In this experiment, we did not intentionally make the specific orientation of the Chinese character different from the arrow angle at the corresponding position. For instance, the Chinese character “up” might be the mask of a vertically upward arrow. This allowed us to compare the differences in the correct recall number when the mask orientation aligned with the target orientation versus when it did not. A significant main effect of mask–target orientation congruency was found (*F*(1, 23) = 6.77, *p* = 0.016, $\eta_{p}^{2}$ = 0.227). Correct recall number was higher when the mask orientation aligned with the target than it did not (reached significance in cue delay of 447 ms condition with *p* = 0.013). The data were further categorized into six levels based on the difference in orientation between the mask and the target: 0°, 45°, 90°, 135°,180°, and Chinese characters without orientation. This categorization aimed to investigate the effect of mask orientation relative to target orientation (Figure 3c). The results indicated that the orientation difference between the mask and the target exhibited a significant main effect (*F*(5, 115) = 7.92, *p* < 0.001, $\eta_{p}^{2}$ = 0.256). Masking effects were significantly stronger in the 45° and 135° conditions than in the condition with Chinese characters without orientation (*p* < 0.001). This pattern of results was consistent across different report-type conditions (see Appendix A for further details).

The high correct recall number in 0°, 90°, and 180° may have been due to the targets being vertical or horizontal under these conditions (but diagonal under the 45° and 135° conditions). We ran an analysis with data in the condition using Chinese characters without orientational information only, and confirmed that the vertical and horizontal targets were recalled better than the diagonal target (*F*(1, 23) = 19.03, *p* < 0.001, $\eta_{p}^{2}$ = 0.453; *p* = 0.003 in cue delay of 0 ms condition, *p* = 0.001 in cue delay of 447 ms condition, and *p* = 0.167 in whole report condition). To avoid being affected by the memory advantages in vertical and horizontal orientations, another 2 (mask type) × 3 (report type) repeated-measures ANOVA was conducted for the diagonal target only. The results consistently demonstrated a stronger masking effect for Chinese orientation characters (*F*(1, 23) = 14.83, *p* = 0.001, $\eta_{p}^{2}$ = 0.392). Further analysis showed that, across all three report types, the correct recall number under the orientational mask condition was significantly lower than under the non-orientational mask condition (with Bonferroni correction: *p* = 0.034 for partial report with cue delay of 0 ms; *p* = 0.013 for partial report with cue delay of 447 ms; *p* = 0.009 for whole report; see Appendix A for more details).

These results again revealed the stronger interference from the orientation semantic mask. The interference observed in conditions when the orientations of the mask and the target were inconsistent strongly support the conclusion that the interference stemmed from the abstract semantic information of orientation rather than specific directional properties. Notably, when the mask and the target shared the same orientation (or inverse orientation at 180°), the correct report number was slightly higher than in other conditions, suggesting that the observed effects may also contain facilitation mechanisms. This will be discussed further in the next section.

## 5. Discussion

Visual representation is a fundamental issue in sensory memory, often investigated using the masking paradigm ([4]; [33]). Across three experiments, we found that in the orientation memory task, masks containing orientation information produced a stronger masking effect than those lacking orientation information. This effect was observed not only between geometric masks with and without orientation semantic information (in Experiment 1 and Experiment 2) but also between Chinese characters with and without orientation information used as masks (Experiment 3). The latter provided particularly strong evidence that orientation information plays a crucial role, because the orientation information carried by Chinese characters was purely semantic and unrelated to their physical features. Furthermore, the masking effect does not result from the specific direction of mask and target but rather from the semantic orientation information. In Experiments 1 and 2, each mask’s specific direction and the target’s direction at the same location differed. In Experiment 3, the analysis also revealed a lower correct recall number in conditions with an orientation mask when the difference between mask and target directions were 45° and 135°. Thus, the interference between the mask and target can be attributed to shared semantic orientation information represented in sensory memory.

A consistent trend emerged from the analysis of data from Experiment 3. Specifically, when the orientation differences between the mask and the target were 0°, 90°, or 180°, the correct recall numbers were higher than in the other two conditions. This can primarily be attributed to the fact that, in these three conditions, the targets were either vertical or horizontal, whereas in the 45° and 135° conditions, the targets were diagonal. Vertical and horizontal orientations hold special significance for the visual system, resulting in better memory performance compared to diagonal orientations. This advantage for vertical and horizontal orientations was indeed observed in the condition without an orientational mask in Experiment 3. However, for diagonal targets only, the Chinese orientation characters still exhibited a stronger masking effect than Chinese characters without orientation information, highlighting the interference of semantic information in sensory memory. The correct recall numbers in the 0° and 180° conditions appeared higher than in the 90° condition, suggesting that the observed results may reflect not only an interference effect but also a facilitation effect. The facilitation effect is plausible, as the similar orientation between the mask and the target in the 0° and 180° conditions may reduce memory impairment caused by confusion between the two sources and enhance memory through repeated presentations. This suggests that the representation of orientational semantic information may exert both facilitative and interfering influences. However, due to the lack of strictly controlled experimental conditions, the presence of a facilitation effect cannot be conclusively determined.

Notably, no partial report effect was observed across the three experiments, consistent with findings from several prior studies. Since Sperling introduced the partial report paradigm to study sensory memory, this effect has become a widely used indicator in sensory memory research ([13]; [34]). In the partial report task (e.g., where the number of rows equals i and the number of columns equals j), the cue is randomly assigned to one of the rows. Participants cannot predict which row the cue will highlight. Researchers typically assume that participants treat all rows equally. When participants correctly report n items in the cued row, it is reasonable to infer that they can also report the same number of items in other rows. Therefore, n is often multiplied by the number of rows (i) to estimate the total number of items remembered by the participants. Sperling found that, under the partial report condition with a short cue delay (such as the 0 ms condition in this study), the number of items remembered by participants was significantly higher than under the long cue delay condition (e.g., the 447 ms condition in this study) or the whole report condition. The latter two conditions are generally close to the capacity of short-term memory. This phenomenon, known as the partial report effect, demonstrates that humans can briefly retain a large number of items, but these memories decay rapidly, with only a few items being consolidated into short-term memory. Although a series of studies have interpreted the partial report effect as evidence of sensory memory, it actually supports the notion that sensory memory capacity is large, rather than proving the existence of sensory memory itself. Even if the partial report effect is not present, this does not imply that sensory memory is absent ([40]; [18]). Sensory memory represents the initial stage of the memory process, defined as a memory stage that occurs within 500 ms. Over time, some of this information will be transferred to short-term or even long-term memory. However, regardless of how much information progresses to later stages, it is initially stored in sensory memory. In this study, the mask completely obscured the target spatially and had a higher contrast against the background than the target, resulting in a strong masking effect that rendered the task challenging. Even without orientation information, participants’ performance in the short cue delay condition did not exceed three items, a number below the typical short-term memory capacity. Thus, the absence of the partial report effect was reasonable. Nonetheless, our focus was on mask type differences. Although even masks lacking semantic information produced a strong masking effect, masks with semantic information further decreased memory performance. This difference in masking effect is best explained by orientation information. Therefore, our findings provide strong evidence that orientation semantic information is encoded in the representation of visual sensory memory.

Extracting semantic information from visual stimuli is a unique and powerful capability of the human visual system. This ability supports visual perception, enabling the recognition of meanings such as relationships, social interactions, functions, and events from stimuli with simple physical features ([1]; [10]; [31]), which are then stored in memory. Recent studies using the mask paradigm have shown that the representation of semantic information already exists at the initial sensory memory stage of visual processing. For example, the sensory memory of Arabic numerals was disrupted not only by other Arabic numerals but also by Chinese capital numerals with different appearances, indicating that the sensory memory representation of specific numerals includes the semantic concept of number ([18]). Our study aligns with this perspective and provides additional evidence that the semantic concept of orientation is also encoded in sensory memory. Moreover, in tasks involving Arabic numerals, participants had to recall the identity of numerals at specific positions, inherently involving semantic processing. In contrast, the current study required participants to remember the specific orientation of stimuli, a task closely tied to physical features. Given the presence of neurons that specifically encode orientation information ([19]; [23]), participants should be able to represent only the physical features of the target. However, our findings revealed that participants’ representations also contained orientation semantic information, suggesting that visual sensory memory may automatically encode and store the semantic content of visual stimuli.

Visual representation is the mental expression of visual stimuli, encompassing not only the physical features detected by the sensory system but also the meaning embedded within these stimuli, which serves essential adaptive functions for humans ([15]; [24]). When interacting with the environment, humans seek not only to survive but also to understand and predict their surroundings. Finding meaning is both a critical driver and a central goal of the cognitive process. Semantic information in visual representation provides a foundation for constructing meaning in visual scenes, and may further help bridge the semantic gap between human vision and communication ([32]; [36]). In addition, to manage the challenge of limited cognitive resources posed by the vast information in the environment, the visual system must integrate information effectively ([37]; [38]). Semantic information in visual representation guides this process, enabling the visual system to integrate semantically related information meaningfully.

The representation of visual sensory memory encompasses both the physical features of the stimuli and their semantic information. For the masking effect, physical features and semantic information may function independently. In Experiments 1 and 2, complete geometric figures were usually used as flash masks, contributing to masking by briefly altering the background brightness. In contrast, the Chinese characters used in Experiment 3 served as pattern masks, potentially exhibiting different effects from flash masks. Nevertheless, we sought to ensure that the masking effects attributable to the physical features of both mask types were as consistent as possible in the same experiment. The observed stable differences supposed that semantic information introduced additional masking effects independent of physical features. These two types of information may be represented as independent. For instance, physical features can be represented in the form of snapshots in visual sensory memory, while semantic information might be processed separately, with the two linked only by their spatial position. Alternatively, the two types of information could form a hierarchical representation, with physical features at a lower level and semantic information at a higher level ([41]). This hierarchical representation not only integrates both types of information but also establishes their relationship. How visual sensory memory represents physical features and semantic information remains an open question, warranting further exploration from the perspectives of psychology, neuroscience, and computational science.

## Figures and Tables

**Figure 1 behavsci-15-00001-f001:**
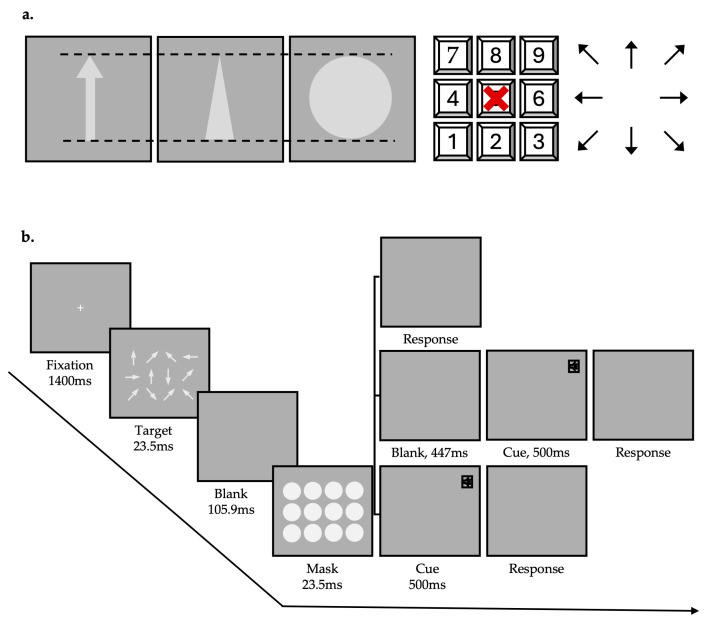
Stimuli and procedure of Experiment 1. (**a**) The left side displays the target and masks used in Experiment 1, while the right side shows the numeric keypad and its mapping relationship to the orientation of arrows. (**b**) Task illustration. The speaker icon indicates a 500 ms tone cue, with three distinct frequencies: high, medium, and low, corresponding to recall requirements for the top, middle, and bottom rows, respectively.

**Figure 2 behavsci-15-00001-f002:**
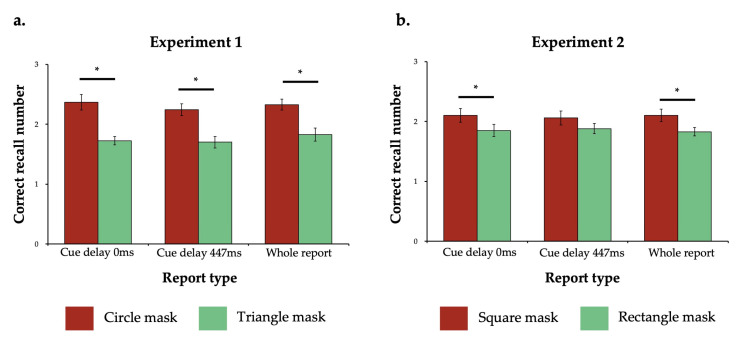
Results of Experiment 1 (**a**) and Experiment 2 (**b**). Correct recall number in Experiment 1. The error bars indicate standard error. The star mark indicates *p* < 0.05 with Bonferroni correction. Note that in the cue delay 447 ms condition in Experiment 2, the *p*-value after correction did not reach significance (*p* = 0.056).

**Figure 3 behavsci-15-00001-f003:**
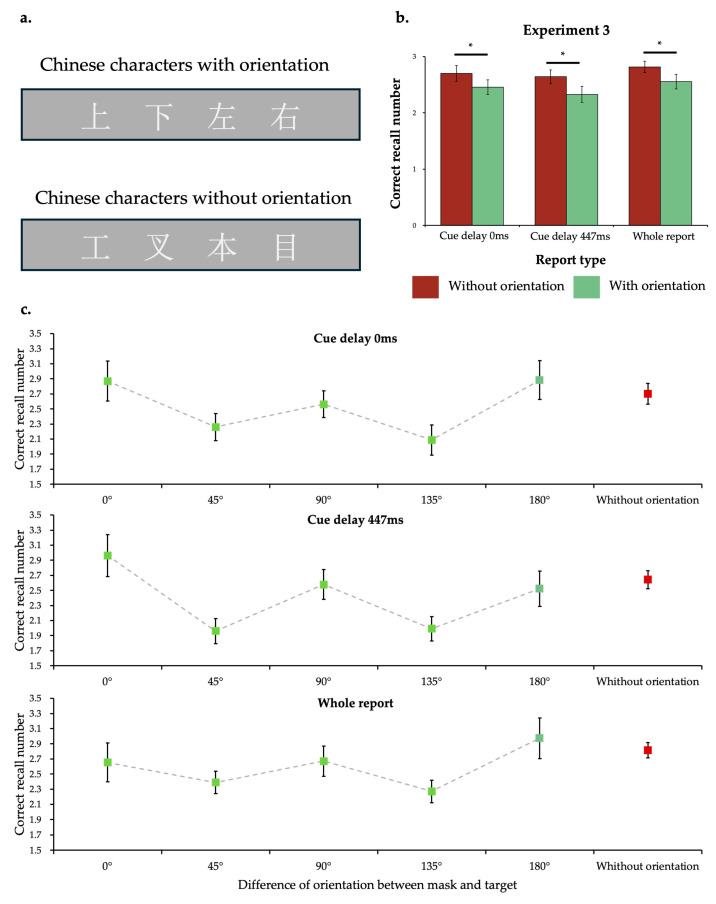
Stimuli and results of Experiment 3. (**a**) Chinese characters used in Experiment 3. The meanings of four characters in the first row are up, down, left, and right. The four characters in the second row have the same number of strokes as the corresponding characters in the first row but do not have any orientation meaning. (**b**) Correct recall number in Experiment 3. The error bars indicate standard error. The star mark indicates *p* < 0.05 with Bonferroni correction. (**c**) The effect of the orientation difference between the mask and the target on the masking effect. The data points represent the mean correct report number under the corresponding conditions, while the error bars indicate the standard error.

## Data Availability

Data available in a publicly accessible repository and could be find from the link: https://osf.io/d2e7n/?view_only=48bd3c8f3f2946bb9f93982148ad9b3a (accessed on 28 October 2024). Appendix A could be found from the same link.

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
