# Peer review of "The Representation of Orientation Semantics in Visual Sensory Memory"

_behavsci, 2024, doi:10.3390/bs15010001_

Round 1
Reviewer 1 Report
Comments and Suggestions for Authors
The authors present an interesting contribution demonstrating that that recall of briefly presented oriented visual stimuli is affected by whether stimuli are masked by oriented or non-oriented masking stimuli. Interestingly, similar effects emerged when the masking stimuli were not visual but semantic (denoting an orientation). The authors interpret this as evidence that sensory memory is affected by semantic information.
The present series of three experimental studies is well-conducted, the analyses are sound, and the results are generally convincing. The findings are highly interesting for the intended audience, the readers of Behavioral Sciences. Hence, I clearly recommend a version of this paper should be published.
My primary concern is with the interpretation of the results as taking place in sensory memory. As far as I know, the latter is assumed to be modality specific. Hence, interference by semantic stimuli (as tested in Experiment 3) may likely not take place in iconic visual memory. I suppose the demonstrated effects rather take place at a later stage, namely propositional encoding and response selection. At this stage, S-S (or S-R) interference could take place when task-relevant propositional representations of imperative stimuli and mask stimuli interfere (or both with response selection).
As the least, it could be tested if the interference of orienting information is (A) a task-irrelevant distraction (as supposed to take place in the modified Stroop paradigm) or (B) an interference of response-related stimulus information (as assumed to take place, e.g., in the priming paradigm). Specifically, the latter would result in fewer errors and possibly faster responses in case of response-congruent stimulus-mask pairs. In case of response-incongruent pairs, both accounts would predict relative error and RT increases.
Minor issues / general remarks
- Were response times also analyzed? Was there an analogous pattern of results (or at least not a systematic trade-off).
- l. 215: "participants recall worse in triangle mask condition than in circle mask condition": I assume this should read rectangle vs. square for Experiment 2?
- There are a couple of typos: "Fugre" (l. 257), "Edperiment" (l. 265), "funcamental" (l. 281), ... Another round of proofreading would be good. Apart from this, language was fine.
Author Response
Thank you for your constructive comments and suggestions, which helped us greatly improve the quality of the manuscript. We have carefully considered each argument and revised the manuscript accordingly. Below, we present our responses to each point and explain in detail how the manuscript has been modified. We have marked the sections of the paper where substantial changes were made in red.
Comments 1: My primary concern is with the interpretation of the results as taking place in sensory memory. As far as I know, the latter is assumed to be modality specific. Hence, interference by semantic stimuli (as tested in Experiment 3) may likely not take place in iconic visual memory. I suppose the demonstrated effects rather take place at a later stage, namely propositional encoding and response selection. At this stage, S-S (or S-R) interference could take place when task-relevant propositional representations of imperative stimuli and mask stimuli interfere (or both with response selection).
Response 1:
In this study, we adopt a concise definition of sensory memory as the memory process that occurs within the first 500 ms. This definition does not subdivide the pre-categorical stage from later stages. Instead, it suggests that semantic information can influence the entire memory process, for example, early traces may also involve the meanings of visual scenes. Based on this definition, we have not further subdivided or discussed which stage semantic interference occurs in. However, when comparing our results with theoretical models that distinguish stages, we agree that semantic interference is more likely to occur at a later stage (propositional encoding and response selection). This is not only because the pre-categorical stage was clearly defined as not containing high-level information, but also because the ISI between the target and mask in our task exceeded the 100 ms threshold typically associated with the pre-categorical stage (not to mention the reaction time). Overall, we suppose that whether interference occurs in the early stages of processing cannot be inferred from our experiments and requires further exploration in future research.
Comment 2: As the least, it could be tested if the interference of orienting information is (A) a task-irrelevant distraction (as supposed to take place in the modified Stroop paradigm) or (B) an interference of response-related stimulus information (as assumed to take place, e.g., in the priming paradigm). Specifically, the latter would result in fewer errors and possibly faster responses in case of response-congruent stimulus-mask pairs. In case of response-incongruent pairs, both accounts would predict relative error and RT increases.
Response 2: Thank you for this insightful suggestion. Unfortunately, in Experiments 1 and 2, we intentionally set the masking and target directions to differ for the same position, resulting in no congruent stimulus-mask pairs. However, such analysis was possible in Experiment 3. Upon splitting the data, we found no significant differences in the number of correct recalls between congruent pairs (e.g., a vertical up arrow paired with the Chinese character "up") and incongruent pairs (e.g., a vertical up arrow paired with Chinese characters "left," "right," or "down"). These results suggest that task-irrelevant distraction may underlie the observed effect. Nonetheless, we observed a tendency for correct recall numbers to be higher in congruent pairs than in incongruent pairs (0 ms: 2.75 vs. 2.42; 447 ms: 2.35 vs. 2.33; whole report: 2.61 vs. 2.54). Since this analysis was conducted only in Experiment 3, we prefer to withhold conclusions about the mechanism until future research provides more direct and comprehensive evidence. We have added this analysis in the results part of Experiment 3 (Line 299 to 305, also in Supplemental Materials online in the OSF) and discussion section (Line 322 to 324).
Comments 3 (Minor issues / general remarks):
- Were response times also analyzed? Was there an analogous pattern of results (or at least not a systematic trade-off).
Response: Thank you for pointing this out. We analyzed response times and did not find significant differences between mask types in all three experiments. Therefore, the main effects in this study should not be attributed to a RT-accuracy trade-off. The response time analysis has been added to the Results sections of the manuscript (Line 183 to 186, Line 235 to 237, and Line 296 to 299. Also in Supplemental Materials online in the OSF).
- l. 215: "participants recall worse in triangle mask condition than in circle mask condition": I assume this should read rectangle vs. square for Experiment 2?
Response: Corrected, thanks.
- There are a couple of typos: "Fugre" (l. 257), "Edperiment" (l. 265), "funcamental" (l. 281), ... Another round of proofreading would be good. Apart from this, language was fine.
Response: Corrected, thanks.
Reviewer 2 Report
Comments and Suggestions for Authors
see attached file

Author Response
Thank you for your constructive comments and suggestions, which helped us greatly improve the quality of the manuscript. We have carefully considered each argument and revised the manuscript accordingly. Below, we present our responses to each point and explain in detail how the manuscript has been modified. We have marked the sections of the paper where substantial changes were made in red.
Comments 1: In my opinion, which is in fact shared with the authors (line 235), experiments 1 and 2 are worthless. They do not prove that ‘semantic’ information is involved, only that physical chages will change the extent of masking in a predictable manner. Much of my commentary below explains why I say this, but in truth, I do not see why the authors bothered to present experiments 1 and 2. I strongly recommend excluding them, rather than struggling to justify their inclusion, which will only irritate the careful reader.
Response 1: Thank you for your suggestion. Indeed, Experiment 3 more thoroughly separated orientation semantic information from physical features, and therefore it provided the strongest evidence. However, after careful consideration, we believe that Experiments 1 and 2 should also be retained in the manuscript for the following reasons: First, Chinese characters are familiar symbols to individuals who have received Chinese education (mostly in China), and people who are not familiar with Chinese may not be able to obtain semantic information from them. In contrast, people with different cultural and educational backgrounds are generally able to recognize geometric figures and have a consistent understanding of the orientation information they may carry. Therefore, experimental evidences from geometric masks are more representative of the universal visual processing mechanism of humans. Considering the generalizability of the experimental conclusions, it is valuable to retain these experiments. Second, providing more parallel evidence from different stimulus materials would help provide support for the universality of the effect. Furthermore, during the course of the study, we did conduct Experiments 1 and 2 before conducting Experiment 3. From a transparency perspective, we believe that reporting them is more or less necessary. Therefore, considering that the editor also suggested to keep Experiment 1 and Experiment 2, we kept these two experiments in the manuscript. However, we modified some statements in the Discussion (Line 313 to 319) to emphasize the importance of Experiment 3 to the conclusion of the whole study. We hope that such modifications were good to the manuscript.
Comments 2: Experiment 3 is highly interesting, well conducted, and can be published, but only with further work. First, the data should be broken down by target angle. For ‘semantic’ interference to be shown, the ‘up’ chinese character should have different effects on near vertical arrow targets than on near horizontal ones. And so forth. Averaging over arrow angle obscures any possibility of understanding the cause of the effect.
Response 2: Thank you for your inspiring suggestions. In Experiments 1 and 2, we intentionally set the masking and target directions to differ for the same position, which made further data segmentation for analysis impossible (Line 122 to 125, and Line 223 to 224). However, this approach confirms that the observed effects in these experiments were not caused by partial conditions where the masking and target directions aligned. Such analysis is possible for Experiment 3. After splitting the data, we found no significant difference in the number of correct recalls between conditions where the masking and target directions were identical and those where they differed. Thus, it is reasonable to infer that the stronger masking effect of Chinese orientation characters arises from conflicts occurring at the conceptual level of orientation semantics. We have added this analysis in the results part of Experiment 3 (Line 299 to 305, also in Supplemental Materials online in the OSF) and discussion section (Line 322 to 324).
Comments 3: Second, a control needs to be run in which the characters are spoken. It may be that the mere presence of ‘Up’ , for example, is enough to distract the subject on some trials, whether the ‘up’ is presented visually or auditorily. Anything the disturbs the subject’s attention will show a reduction in recall, but this need not be due to visual interference. For example, hearing ‘up’ while seeing a left arrow may confuse or distract a subject, especially after many fatiguing trials, and this effect may equal the (rather small) magnitude of the ‘memory’ loss seen in expt. 3. Distraction would predict that the interference was independent of cue delay or report method, which indeed seems to be the case.
Response 3: Auditory masking presents a promising approach, as it can entirely eliminate the influence of physical visual features. We were inclined to conduct this experiment, anticipating that auditory directional words would introduce greater interference. However, we also believe the cross-channel masking issue explored in the auditory experiment extends somewhat beyond the primary scope of this study. Considering the limited time available for this round of revisions, we regret to postpone this experiment for the current study and will incorporate it into a subsequent investigation. We sincerely appreciate your suggestions.
Comments 4 (Small points):
-The term 'semantic’ needs clarification in the Abstract, as the word encompasses many levels of meaning including purely verbal. Otherwise, the Abstract is clear and informative.
Response: Thank you. We have included a sentence in the Abstract to clarify the meaning of "semantic" as it pertains to the current study.
-The term ‘sensory memory’ has been used in the past for low-level information conveyed by the visual sensory apparatus, and for example was been attributed to the retinal after-image by Sakitt and Long. However, the research question here involves a higher-level, perceptual representation, one of orientation. So the direct question, whether ‘sensory’ memory is ‘semantic’, make little sense historically. One way around this is to use Sperling’s (1960) term ‘visual information store’ (VIS), renamed ‘the icon’ by Neisser, with the remark that the phsysiological substrate of VIS includes, but is not limited to, visual sensations. Or the authors may just use the term ‘sensory memory’ to mean simply a visual memory that is limited to the first 500 ms, but if so, they should clarify this terminology.
Response: In this study, we adopt a concise definition of sensory memory as the memory process that occurs within the first 500 ms. This definition avoids the historical, rigid view that sensory memory contains no high-level information and aligns with certain modern theories. In the revised manuscript, we clarify this terminology in both the introduction (Line 55) and discussion sections (Line 347 to 351).
-Line 79: ‘By comparing the interference caused by the two types of masks, we can determine whether the representation contains semantic information about orientation.’ This logic is missing a premise, namely, that the low-level featural information is NOT responsible for interference (which must then be ‘semantic’) . Suppose to the contrary that the chinese characters, triangles, etc, contained the same orientated features (e.g., diagonals and uprights) as the arrow targets. Then the best that could be concluded is that physically similar features are better masks than dissimilar ones. ‘Semantic’ similarity would not be implied. Indeed, the authors eventually admit this themselves.
Response: Thank you for highlighting the weakness in the logical argument. In the revised manuscript, we have emphasized the similarity of physical features between masks (Line 85, 88 to 89). Differences in masking effects can be attributed solely to semantic information when the physical features are consistent.
-The authors also used circles and squares as masks. However, such masks, as portrayed in Fig. 1b, simply act as a temporary increase of background luminance, what has been termed a ‘flash mask’. It has been known since the 1960s that flash masks are less effective than pattern masks. What would be required is some evidence that the contours around the circle and square masks are interfering, not just the change in luminance level.
Response: The masking effects of physical features may have multiple underlying causes. In Experiments 1 and 2, geometric figures were primarily used as flash masks, facilitating masking by briefly altering background brightness. Conversely, the Chinese characters employed in Experiment 3 served as pattern masks. We sought to ensure that the masking effects attributable to the physical features of both mask types were as consistent as possible in the same experiment. As a result, the observed stable differences support the role of semantic information in masking. Additional sentences addressing this were incorporated into the discussion section (Line 394 to 402). We speculate that even when physical features act through contours, semantic interference may persist. However, the current experiments do not provide direct evidence to confirm this hypothesis, necessitating further investigation in future studies, such as those involving virtual contours.
-Fig. 2 appears to be drawn incorrectly. The text states that there were more errors with the triangle mask than the circle mask, but the figure (in my pdf) shows the opposite. I suggest using dotted bars for one mask and solid bars for the other, so that, even without color, the Figure can be correctly interpreted.
Response: We apologize that the legends in Figure 2a and Figure 2b were reversed. After careful review, we ensured that the original data (the data uploaded to OSF) were correct and that the analysis and textual presentation were correct. We have corrected the legends in the new version.
-Line 220; I do not follow this reasoning. From fig. 1, it seems that the arrows are pointed vertically, so a vertical rectangle is more similar physically to a vertical arrow than it is to a square, so it sould have a greater masking effect due to physical similarity. Suppose now that the vertical rectangle and the square have equal ‘flash’ masking effects on horizontal or near-horizontal arrow targets. Then, on average, the rectangle will mask more than the square. What is needed is a break-down of the data by arrow angle, to show that the rectangle is more effective at every orientation (not just near, or at, vertical). In this way, the authors could show that the ’semantic’ content of the rectangle was a relevant factor.
Response: Thank you for your suggestion. When we state that "other appearance features of rectangle and square are highly similar", we primarily mean that both shapes have four sides, four right angles and closely related to each other geometrically. The use of rectangles and squares increases physical similarity compared to using triangles and circles. However, further analysis would be beneficial. As noted earlier, in Experiments 1 and 2, we intentionally set the masking and target directions to differ for the same position. Consequently, there were no positions with vertical up/down arrows paired with vertical rectangles, rendering further data segmentation for analysis impossible. However, in Experiment 3, this analysis was conducted, and no significant differences were observed between conditions where the masking and target directions were identical and those where they differed. Therefore, it might be reasonable to conclude that the "semantic" content was a relevant factor, particularly for Chinese characters. Future research could further explore the varying effects of semantic information at different levels, such as orientation information with same direction comparing with orientation information with different direction.
-Line 265. ‘Edperiment’ should be ‘Experiment’.
Response: Corrected, thanks.
Reviewer 3 Report
Comments and Suggestions for Authors
This paper examined whether semantic information, specifically orientation, is involved in visual sensory memory representations. Three experiments were conducted in order. The methodology is solid and the analysis is complete. The paper might be suitable for publication if the authors will attend several small issues and revise the text.
1. Line 48: “Although the the mask”
2. Line 96: how was the visual acuity determined?
3. In Figure 2a, experiment 1 results show that when using the “triangle mask” the correct recall number is higher than that of the “circle mask”, is this correct?? Similarly for 2b.
4. Line 215: “…participants recall worse in triangle mask condition than in circle mask condition…”?? experiment 2 does not have triangle and circle conditions, correct?
5. Figure 3b legend? What do red and green bars mean?
Author Response
Thank you for your constructive comments and suggestions, which helped us greatly improve the quality of the manuscript. We have carefully considered each argument and revised the manuscript accordingly. Below, we present our responses to each point and explain in detail how the manuscript has been modified. We have marked the sections of the paper where substantial changes were made in red.
Comments 1 (Minor):
-Line 48: “Although the the mask”
Response: Corrected, thanks.
-Line 96: how was the visual acuity determined?
Response: The normal or corrected-to-normal visual acuity is determined as follow: the naked eye or with glasses is above 4.8 as tested by international standard vision chart. We have added the statement (Line 105 to 106).
-In Figure 2a, experiment 1 results show that when using the “triangle mask” the correct recall number is higher than that of the “circle mask”, is this correct? Similarly for 2b.
Response: We apologize that the legends in Figure 2a and Figure 2b were reversed. After careful review, we ensured that the original data (the data uploaded to OSF) were correct and that the analysis and textual presentation were correct. We have corrected the legends in the new version.
-Line 215: “…participants recall worse in triangle mask condition than in circle mask condition…”?? experiment 2 does not have triangle and circle conditions, correct?
Response: Thank you very much for pointing out this error. We have corrected it (Line 232 to 233)
-Figure 3b legend? What do red and green bars mean?
Response: We apologize that the legend in Figure 3b was missing. This figure in the new version of manuscript has been corrected.
Round 2
Reviewer 1 Report
Comments and Suggestions for Authors
The authors were very responsive and have adequately addressed all points raised in the previous round. In particular the insertion of the paragraph defining what is actually meant by "sensory memory" has helped clarify what is actually subsumed here.
In my eyes, the manuscript is ready for publication. I am sure readers will appreciate this informative contribution to the literature of visual processing.
Author Response
Comments1: The authors were very responsive and have adequately addressed all points raised in the previous round. In particular the insertion of the paragraph defining what is actually meant by "sensory memory" has helped clarify what is actually subsumed here. In my eyes, the manuscript is ready for publication. I am sure readers will appreciate this informative contribution to the literature of visual processing.
Response1: Thank you for your approval of the revised manuscript.
Reviewer 2 Report
Comments and Suggestions for Authors
I continue to maintain that expts 1 and 2 should be excluded. It is not news that mask orientation can affect target identification at a sensory level. For example, vertical line masks reduce acuity for seeing vertical line targets, but horizontal line masks do not.
Expt 3 deserves to be published, as novel, because the semantic content of the characters is shown to be relevant. However I would prefer to see a graph of masking extent versus mask orientation (relative to target orientation) to verify that sensory factors are not sufficient to explain the (semantic) orientation effect, as I explained in the original review. I am unsure why the authors did not plot the data in this way, but only plotted an average across all orientations (in Fig. 3).
Author Response
Comments 1: I continue to maintain that expts 1 and 2 should be excluded. It is not news that mask orientation can affect target identification at a sensory level. For example, vertical line masks reduce acuity for seeing vertical line targets, but horizontal line masks do not.
Response 1: Thank you for your feedback. As mentioned in the previous revision, we prefer to retain both Experiment 1 and Experiment 2 to ensure a comprehensive and transparent study, in line with the editor's suggestions.
Comments 2: Expt 3 deserves to be published, as novel, because the semantic content of the characters is shown to be relevant. However, I would prefer to see a graph of masking extent versus mask orientation (relative to target orientation) to verify that sensory factors are not sufficient to explain the (semantic) orientation effect, as I explained in the original review. I am unsure why the authors did not plot the data in this way, but only plotted an average across all orientations (in Fig. 3).
Response 2: Thank you for your inspiring suggestions. We divided the mask conditions into those consistent or inconsistent with the target orientation and found a significant difference between them. (We apologize for reporting no significant difference in the previous version of the revision; this was due to an error caused by discrepancies in the encoding of the mask and target orientations in the analysis program. This error has been corrected in the current version). Specifically, when the mask and target orientations were consistent, the correct recall number was higher compared to when they were inconsistent, indicating a facilitation effect of consistency between the mask and target.
To further investigate this effect, the data were categorized into six levels based on the orientation difference between the mask and the target: 0°, 45°, 90°, 135°, 180°, and without orientation. The results revealed that the difference in orientation significantly influenced the masking effect. The weakest masking occurred under the 0° and 180° conditions, followed by the 90° condition. The strongest masking effects were observed under the 45° and 135° conditions, which showed significant differences from the non-orientational mask condition.
The reduced masking effect in 0°, 90° and 180° conditions may be due to the targets were vertical or horizontal under these conditions (but diagonal under the 45° and 135° conditions). We confirmed this by analysing the data in no-orientation conditions. Moreover, we compared the effects of orientation and no-orientation masks by focusing solely on diagonal targets. The results consistently demonstrated a stronger masking effect when the mask was an orientational character. These findings further support the conclusion that the semantic information conveyed by the mask's orientation creates conflict and interferes with target memory. We also considered the possibility that the mask in the 0° and 180° conditions might produce a facilitation effect. However, due to the lack of strictly controlled experimental conditions, it remains an open question whether the representation of semantic information exhibits both facilitative and interfering effects.
The above analysis and discussion have been incorporated into the new manuscript (in red in Line 315 to 350, Line 369 to 388), and Figure 3 has been updated. The new analysis results were also included in the supplementary material of this revision.